# Non-Alcoholic Fatty Liver Disease (NAFLD) Management in the Community

**DOI:** 10.3390/ijms26062758

**Published:** 2025-03-19

**Authors:** Yongsoo Park, Kyung Soo Ko, Byoung Doo Rhee

**Affiliations:** Department of Internal Medicine, Sanggye Paik Hospital, College of Medicine, Inje University, 1342 Dongil-ro, Nowon-gu, Seoul 01757, Republic of Korea; kskomd@paik.ac.kr (K.S.K.); bdrhee@paik.ac.kr (B.D.R.)

**Keywords:** non-alcoholic fatty liver disease, metabolic syndrome, diabetes, liver fibrosis, community

## Abstract

Non-alcoholic fatty liver disease (NAFLD) has frequently been associated with obesity, type 2 diabetes (T2D), and dyslipidemia, all of which are shared by increased insulin resistance. It has become the most common liver disorder in Korea as well as in developed countries and is therefore associated with an increased health burden of morbidity and mortality. It has an association with T2D, and T2D increases the risk of cirrhosis and related complications. NAFLD encompasses a disease continuum from simple steatosis to non-alcoholic steatohepatitis which is characterized by faster fibrosis progression. Although its liver-related complication is estimated to be, at most, 10%, it will be a leading cause of cirrhosis and hepatocellular carcinoma soon in Korea. Although the main causes of death in people with NAFLD are cardiovascular disease and extra-hepatic malignancy, advanced liver fibrosis is a key prognostic marker for liver-related outcomes and can be assessed with combinations of non-invasive tests in the community. A number of components of metabolic syndrome involved could be another important prognostic information of NAFLD assessed easily in the routine care of the community. There is a few approved therapies for NAFLD, although several drugs, including antioxidants, attract practitioners’ attention. Because of the modest effect of the present therapeutics, let alone complex pathophysiology and substantial heterogeneity of disease phenotypes, combination treatment is a viable option for many patients with NAFLD in the Korean community. Comprehensive approach taking healthy lifestyle and weight reduction into account remain a mainstay to the prevention and treatment of NAFLD.

## 1. Introduction

As type 2 diabetes (T2D) increases the concentration of glucose in the blood of adult population, kidney and nerve damage as well as retinal complications will naturally ensue as its consequences. It can also cause serious cardiovascular complications (CVDs) such as peripheral vascular disease, stroke, angina, and/or myocardial infarction. In the long run, these all contribute to increased mortality and morbidity of diabetes patients and reduce their life expectancy. It was not until the 1960s, along with rapid modernization as well as social and economic growth with improved medical utilization patterns, that the number of diabetes patients was found to increase explosively. Owing to shifts in the dietary composition as well as excessive nutrient intake, there has been an aggregation of CV risk factors in the general population, suggesting involvement of metabolic syndrome (MS). From the epidemiological data implemented in Yonchon, Gyonggi-do in 1990, we now know that 8 out of 100 adults in Korea have diabetes [1]. A consecutive follow up study, performed 2 years later at the same location, proved that diabetes condition is expected to increase in the future [2]. Therefore, a proper management plan at the community level should be employed, knowing that diabetes is an important health issue in Korea.

Non-alcoholic fatty liver disease (NAFLD), commonly co-associated with T2D, has not recognized as a significant comorbidity. However, it is a chronic metabolic disease that reduces the quality of life for patients with T2D in the community [3]. NAFLD is the liver component of a cluster of conditions that are associated with MS. NAFLD is known to have a bidirectional association with components of MS [4]. Although Koreans have a higher fraction of NAFLD patients who are lean, their illness has the same or worse prognosis than for those who are obese. It is a disease with increased phlegm, which is common in the community, especially in Korea [5]. Over the past four decades, NAFLD has become the most common chronic liver disorder (with a global prevalence of around 25% of the adult population) [4]. As around 10% of people with NAFLD develop liver-related complications, a crucial challenge is to identify those who are at the highest risk for these complications among the people affected by NAFLD. Due to its high prevalence, NAFLD is now the most rapidly increasing cause of liver-related morbidity and mortality and is emerging as an important cause of end-stage liver disease, hepatocellular carcinoma, and liver transplantation with higher medical costs [4,6,7,8]. Gradually, nonalcoholic steato-hepatitis (NASH) and cirrhosis have also emerged as causes for increased health care costs. Among community inhabitants in their 30s to 50s, the overall death rate due to chronic liver diseases, excluding hepatocellular carcinoma, ranks 2nd to 4th in Korea [5]. Therefore, NAFLD is increasingly recognized as a significant health threat and an emerging public health concern. The number of patients with NAFLD is very high, and its complications contribute to substantial mortality, further exacerbated by the high healthcare costs. Despite the growing concern, NAFLD is underappreciated as an important chronic disease, and there are few national strategies or policies for NAFLD [5].

The prevalence of lean NAFLD worldwide appears to be rising at a similar rate to obese NAFLD [9]. Patients with lean NAFLD have a higher mortality rate from liver-related causes than obese NAFLD patients, because they lack obesity and may not have typical MS risk factors associated with insulin resistance (IR). These make lean NAFLD more likely to be unnoticed or be diagnosed late, which, in turn, makes the progression and outcomes worse than obese NAFLD. To find lean NAFLD patients early, genetic markers associated with lean NAFLD have been studied [10]. It is important to be aware of these genetic causes of lean NAFLD as some are treatable. Future research will likely include improvements in diagnostics.

NAFLD is defined by the presence of steatosis in more than 5% of hepatocytes in association with metabolic risk factors (particularly, obesity, and T2D) and in the absence of excessive alcohol consumption or other chronic liver diseases [7,11]. NAFLD is a diagnosis of exclusion and there is a debate about the limitations of this term. In 2020, an international panel of experts proposed the concept of metabolic dysfunction-associated fatty liver disease (MAFLD) to highlight the contribution of cardiometabolic risk factors to the development and progression of liver disease [12]. Therefore, we will use the term ‘MAFLD’ rather than friendly ‘NAFLD’. MAFLD encompasses a disease continuum that includes steatosis with or without mild inflammation and a necroinflammatory subtype (MASH), which is additionally characterized by the presence of hepatocellular injury (hepatocyte ballooning). However, the prime pathogenesis of disease can vary substantially among patients with MAFLD, which could be crucial hurdles to be managed appropriately in the community. Furthermore, disease progression and response to treatment are heterogeneous. Information about disease activity and, in particular, the extent of liver fibrosis are necessary to assess the severity of liver disease and provide prognostic information. Growing knowledge from metabolomics, genomics, and other areas will enable disease phenotyping and facilitate potential disease stratification in the future.

Diabetes and MAFLD are known to be the conditions that often coexist in the community. Research on the cause and clinical significance of this correlation is rare. The rise in diabetes and MAFLD since the late 1960s might result in part from the excessive nutritional intake. However, it might also be related to shifts in the dietary composition taking tremendous changes in the basic nutrient composition of diet into account. According to the Korean Annual National Health and Nutrition Examination Survey, carbohydrate intake has reduced gradually, whereas lipid intake has increased suggesting a role of dyslipidemia in this epidemiologic shift [13]. Generally, as the content of liver fat increases, the likelihood of lipid oxidation also increases accordingly. It has been known that in the circumstances of increased whole-body lipid oxidation, the levels of endogenous glucose production increase and those of glucose disposal decrease [14]. Excess lipids might also induce glucose intolerance through an inhibitory effect on glucose storage [15]. According to a calorimetric experiment report, severe MAFLD patients exhibited a more dramatic decrease in insulin-mediated suppression of whole-body lipid oxidation, implicating deepened IR. This resulted in a smaller reduction in blood β-hydroxy butyrates (ketones) [16]. Likewise, a more pronounced decrease in insulin-mediated suppression of glycerol production was also found in severe MAFLD. Therefore, a lesser degree of insulin-mediated suppression of glucose production as a whole occurs as the level of plasma nonesterified fatty acids (FA) increases, which might be considered as a primary driver of MAFLD.

In around half of people with cirrhosis, a severe complication of MAFLD abnormalities in glucose tolerance are observed. According to one report, 15–20% of patients with liver cirrhosis develop diabetes within 5 years [5]. Detection of T2D among patients with MAFLD or MASH is important in part because improved glycemic control may improve MAFLD. Although there is no difference in the short-term prognosis between cirrhosis patients with and without diabetes, patients with diabetes have a worse long-term prognosis. It is known that liver failure occurs frequently in patients with diabetes. Importantly, advanced liver fibrosis should be counted as a key prognostic marker for liver-related outcomes and can be assessed with combinations of non-invasive tests in the community. In the present review, we would first like to present an overviewing synopsis explaining pathogenesis, pathways of MAFLD progression, along with diagnostic modalities and methods for assessing disease severity, especially when combined with fibrosis. Then, we explore viable prevention and management strategies at the community level. Ultimately, we emphasize the importance of recognizing MAFLD as an important chronic disease and suggest some strategies and policies to overcome it at the community level.

## 2. Pathogenesis in General

The principal mechanism for MAFLD development is overnutrition, which causes not only expansion of adipose tissue but also accumulation of ectopic fat in the liver. Meanwhile, macrophage infiltration of the visceral adipose tissue creates a proinflammatory state that promotes IR [17]. Inappropriate lipolysis under the influence of IR results in unabated delivery of fatty acids to the liver, which, along with increased de novo lipogenesis (DNL), overwhelms its metabolic capacity. The imbalance in lipid metabolism leads to the formation of lipotoxic lipids that contribute to cellular stress (oxidative stress (OS) and endoplasmic reticulum (ER) stress), inflammasome activation and apoptotic cell death, and stimulation of inflammation, tissue regeneration, and fibrogenesis [7,18,19]. Inflammatory and profibrogenic macrophages are implicated in the progression of liver fibrosis and might also have a role in chronic inflammatory processes in other tissues as well [20] (Figure 1).

These pathogenic pathways of MAFLD are influenced by multiple metabolic, genetic, and microbiome-related factors that are not understood completely yet. MAFLD has a heritable component with variable penetrance estimates ranging from 20 to 70% [7]. A single-nucleotide polymorphism (SNP) in the patatin-like phospholipase domain-containing 3 (*PNPLA3*) gene, is the best characterized genetic variant associated with susceptibility to MAFLD, the importance of which has not yet been assessed properly in the Korean community. Utilizing genetic markers found in genome-wide association studies (GWAS), functional studies in rodents, and epigenetic markers, researchers tried to predict the development of MAFLD and MASH. In an ethnically diverse cohort, one group replicated several key genetic variants for MAFLD and showed the utility of genetic risk stratification (GRS) for MAFLD risk prediction [21]. Studies have identified *PNPLA3*, transmembrane 6 superfamily member 2 (*TM6SF2*) and apolipoprotein B gene mutation, and others were thought to be candidates for lean MAFLD risk markers [10]. However, all of known genetic variants account for a small proportion (10–20%) of overall heritability, although this proportion varies across populations. These genes might influence multiple traits such as those related to IR with divergent effects on MAFLD and cardiovascular comorbidities. Several genetic risk variants of MAFLD show a synergistic interaction with those of obesity. In addition, crosstalk between the liver and other organs (particularly adipose tissue and the gut) via FGF19 and FGF21 might also contribute to metabolic dysregulation and inflammation in MAFLD [18,19,20,22,23,24] (Figure 1). Alterations in gut microbiota are seen in patients with MAFLD, and some data suggest that there is a fecal-microbiome signature associated with advanced fibrosis [25,26,27]. However, confirmation of these bacterial signatures in different geographical regions controlling for environmental factors is required to determine their clinical significance and use for diagnostic purposes. Gut microbiota plays a role in the pathophysiology of MAFLD through the gut–liver axis. Microbiome-based therapies (probiotics, prebiotics, synbiotic, fecal microbiota transfer, polyphenols, specific diets, and exercise interventions) have been found to modify gut microbiota signatures and improve MAFLD outcomes. Apart from probiotics that have already been tested in human RCTs, most of these potential therapeutics have been studied in animals. Their efficacy still warrants confirmation in humans [25].

## 3. Molecular Pathways of MAFLD Progression

In around 15% of MAFLD patients, simple steatosis can evolve into MASH, a mixture of inflammation, hepatocellular injury, and fibrosis, often culminating in cirrhosis and even hepatocellular cancer [28]. Although the precise molecular mechanism underlying MAFLD progression is not completely understood, its pathogenesis on the whole has been assumed by the ‘double-hit hypothesis’ [29]. The first hit includes lipid accumulation in the liver, followed by a second hit in which proinflammatory mediators induce inflammation, hepatocellular injury, and fibrosis. Nowadays, a more complex model assumed by ‘multiple parallel hits hypotheses’ suggests that FAs and their metabolites might be truly lipotoxic and they might contribute to MAFLD progression [30,31]. It is certain that in MAFLD patients, IR leads to hepatic steatosis via multiple mechanisms [32]. Not only is DNL increased, but FAs are also excessively taken up into the liver from the serum. Furthermore, a decrease in mitochondrial FA oxidation and secretion of very-low-density lipoproteins (VLDL) has been reported. It turns out that triglyceride (TG) accumulation in the cytoplasm of hepatocytes, as the hallmark of MAFLD, arises from an imbalance between lipid acquisition (FA uptake and DNL) and removal (mitochondrial FA oxidation and export) and accompanies multiple pathophysiological mechanisms in MASH [29].

The first process called DNL includes de novo synthesis of FAs through a continuing process of polymerization in the hepatocyte in which glucose is converted to acetyl-CoA by glycolysis and the oxidation of pyruvate in the beginning. Then, acetyl-CoA carboxylase (ACC) converts acetyl-CoA into malonyl-CoA. Finally, FA synthase (FAS) catalyzes the formation of palmitic acid from malonyl-CoA and acetyl-CoA. Depending on the metabolic needs, FAs are then processed to TGs and stored or rapidly metabolized. On the whole, the rate of DNL is regulated primarily at the transcriptional level. Several nuclear transcription factors such as liver X receptor α (LXRα), sterol regulatory element-binding protein 1c (SREBP1c), carbohydrate-responsive element-binding protein (ChREBP), farnesoid X receptor (FXR), and several enzymes (*ACC*, FAS, and steroyl CoA desaturase 1 (SCD1)) are involved. After the ingestion of meals, rising plasma glucose and insulin levels promote lipogenesis through the activation of ChREBP and SREBP1c, respectively [29]. In humans, MAFLD has been associated with increased hepatic expression of several genes involved in DNL [29,33].

Oxidation of FAs occurs in the mitochondria, peroxisomes, and the ER. It facilitates the degradation of activated FAs to acetyl-CoA. Activated long chain fatty acids (LCFAs) are shuttled across the membrane via carnitine palmitoyltransferase-1 (CPT1). Malonyl-CoA, an important intermediate of DNL, is an inhibitor of CPT1. In FA overload conditions, such as MAFLD and diabetes, cytochrome P450 (CYP4A)-dependent ω-oxidation of LCFAs occurs in the ER and induces ROS and lipid peroxidation. During the process of β-oxidation, electrons are indirectly donated to the electron transport chain (ETC) to drive ATP synthesis. Acetyl-CoA can be further processed via the tricarboxylic acid (TCA) cycle, or in the case of FA abundance, be converted into ketones. PPARα and insulin signaling are involved in the regulation of FA oxidation and the formation of ketones. In the liver, PPARα plays a pivotal role in FA metabolism by upregulating the expression of numerous genes involved in mitochondrial and peroxisome FA oxidation [33,34]. Therefore, the activation of PPARα might prevent and decrease hepatic fat storage.

## 4. Diagnosis and Assessment of Disease Severity

MAFLD is frequently diagnosed by imaging, although it can be inferred from clinical risk scores such as fatty liver index (FLI) or identified histologically. In routine practice, the most commonly used test is abdominal ultrasonography (US) [35,36]. However, abdominal US has two important limitations: insensitivity and misdiagnosis. Advanced fibrosis can coarsen hepatic echotexture, blur the vascular pattern, and lead to misdiagnosis. The sensitivity of US is low when steatosis is mild (<30%). Instead, MRI-based measurements of hepatic steatosis can detect as little as 5% fat and are sensitive to dynamic change, but they are often used in the research setting rather than in routine clinical practice in the community.

Detection of T2D among patients with MAFLD or MASH is important because improved glycemic control may improve MAFLD. MAFLD patients with T2D is associated with a more than two times increased risk of advanced fibrosis, cirrhosis-related complications, and liver disease mortality [7]. Obesity, dyslipidemia, and hypertension are also associated with an increased risk of severe liver disease, although the effect sizes are smaller than for T2D [37,38]. Therefore, a number of components of MS involved should be counted at the first visit and could be used as an important prognostic information of MAFLD assessed easily in the routine care of MAFLD (Figure 2). Patients with MAFLD who are older than 65 years also have a higher prevalence of advanced fibrosis. A variant of the *PNPLA3* gene may be associated with MAFLD histological severity and development of hepatocellular carcinoma as well as liver-related and all-cause mortality. Recently, the *PNPLA3* genotypes in addition to diabetes status were found to identify patients at a higher risk of cirrhosis among those at an indeterminate risk of MAFLD in two Caucasian cohorts [39], the clinical role of which has not been reproduced in a Korean biopsy-proven MAFLD cohort [40].

## 5. Non-Invasive Tests of Disease Severity

Liver enzymes (alanine aminotransferase (ALT) and aspartate aminotransferase (AST)) may be a first step to assess and monitor patients with liver diseases. However, serum liver enzyme concentrations can be normal in more than half of patients with MAFLD and correlate poorly with the histological severity [41]. Traditionally, liver biopsy was used to characterize and quantify histological features of steatosis, inflammation, hepatocyte ballooning, and fibrosis. However, this invasive procedure is not suitable for widespread use to assess progression or response to therapy in the community. In addition to its risk and cost, liver biopsy is prone to sampling bias. Moreover, subjective variability in histological assessment is also documented. As fibrosis has consistently emerged as the most crucial histologic feature predicting clinical events, researchers have developed several non-invasive tests estimating fibrosis. Simple fibrosis scores, such as the NAFLD fibrosis score (NFS), Fibrosis-4 (FIB-4) index and AST-to-platelet ratio index (APRI) composed of demographic, clinical, and routine laboratory parameters have been developed and are ready to be applied in the community. These are inexpensive to use and can be applied easily in the Korean community [42,43]. Although the overall sensitivity of these scores is assumed not to be high, they have high negative predictive values to exclude advanced liver fibrosis. Patients with low fibrosis scores are also at a low risk of developing liver-related complications.

Another way to estimate liver fibrosis in patients with MAFLD is to measure liver stiffness by US-based and MR elastography [44]. Transient elastography (Fibro-scan) has been most extensively evaluated, is widely available, and can therefore be used in the community. It is also possible to estimate hepatic steatosis by measuring the controlled attenuation parameter at the same time. The liver stiffness measurement (LSM) also correlates with the future risk of hepatocellular carcinoma and cirrhotic complications.

In the community, we may start with inexpensive simple fibrosis scores (NFS or FIB-4) as a first step to identify individuals at low risk of advanced fibrosis who can be managed in the community. A number of components of MS involved could be another prognostic marker of MAFLD assessed easily in the routine care in the community. Individuals with high risk of advanced fibrosis require additional assessment with Fibro-scan or might require referral to university hospitals (secondary care) for investigation of liver disease or management of advanced fibrosis. Patients without advanced fibrosis at initial assessment might require ongoing monitoring to identify progressive liver disease and retesting 3 years after initial assessment (Figure 2) [45].

## 6. Comorbid Conditions

Although the risk of liver disease progression is extensive, the main cause of death in patients with MAFLD is CVDs, followed by extrahepatic malignancy such as colorectal or breast cancer [7,28]. These might result from increased cardiometabolic risk factors that are shared in MAFLD and CVDs. However, it is not known to what extent MAFLD has a direct causative role in the development of CVDs. In addition to the characteristic proatherogenic lipid profile, the bidirectional relationship between MAFLD and some MS features (T2D and hypertension) is one mechanism by which MAFLD might augment cardiovascular risk [46,47]. Patients with MAFLD have a 1.9 times higher risk of incident cancers than the general population, particularly cancers involving the liver, gastrointestinal tract and uterus [48]. It might be driven by the association of MAFLD with visceral adiposity and chronic low-grade inflammation, but this mechanism has not yet been determined.

## 7. Prevention and Management of NAFLD

Along with new strategies in the diagnosis, risk stratification, and management of MAFLD, a few studies have succeeded in evaluating primary prevention of MAFLD, which can be organized and managed easily in the community. It turns out that improved diet quality [49] and sustained physical activity [50] reduced the risk of developing MAFLD, even among individuals with high genetic risk. As a first start, clinicians can manage to implement comprehensive programs to promote and coordinate lifestyle interventions with dietary modification and exercise, accompanied by management of metabolic comorbidities. The team approaches to fulfill these activities can be easily organized and accessed at the community level via the holistic medicine pursuing multidisciplinary approach. Having various non-invasive tests to diagnose MAFLD and liver fibrosis at hand, to screen for MAFLD and MAFLD with fibrosis can be amenable, particularly when patients participate in secondary prevention programs for T2D or MS. However, data are inconsistent, partly reflecting the paucity of available effective therapeutic interventional measures [51]. Irrespective of the possibilities of over-diagnosing MAFLD, once MAFLD is diagnosed applying simple clinical indices or abdominal US in the community, practitioners recommend risk stratification by assessing the presence of advanced fibrosis or cirrhosis, and evaluating cardiovascular risk and comorbid illnesses (Figure 2).

It is certain that those patients with T2D have a high prevalence of MAFLD (40–70%) and are more likely to develop advanced fibrosis, cirrhosis, and hepatocellular carcinoma [52]. Multi-morbidity and polypharmacy are commonly found in patients with T2D and MAFLD in several community hospitals of Korea, highlighting a need for multidisciplinary management to address the complicated health care needs [53]. In university diabetes clinics, the prevalence of advanced fibrosis among patients with MAFLD is around 2–4 times higher than in the community [54,55]. Because of the increasing need to assess MAFLD and liver fibrosis that should be incorporated into the routine care of patients with T2D, the American Diabetes Association now recommends that “Patients with T2D and elevated liver enzymes (ALT) or fatty liver on US should be evaluated for the presence of NASH and liver fibrosis” [56]. However, since ALT measurements are inaccurate and are within the normal range in most people with T2D and MAFLD, many patients with clinically significant liver disease will not be diagnosed. Alternatively, we can rephrase them into a better recommendation implicating that “Patients with T2D with elevated FLI or fatty liver on US should be evaluated for the presence of MASH and liver fibrosis applying simple fibrosis scores”.

## 8. Management of MAFLD in the Community

Although the liver-related burden of MASH is substantial and increasing, CVDs and malignancies are the leading causes of death in people with MAFLD [4,46,47,48,54]. Therefore, the management of MASH should adopt a holistic approach that strives to minimize CV risk and to reduce inducers of steatosis and systemic inflammation. Central obesity is an important driver of disease through the promotion of IR and proinflammatory signaling. Although the nutrient content and proportion of the fat in diet is important, weight loss inducing more than 5–7% of present body weight reduces hepatic fat content and MASH, while weight loss in excess of 10% brings about even fibrosis reduction in a large proportion of people, irrespective of the method of weight loss [57]. However, sustained weight loss is challenging because it requires a transformation of behavior patterns. It is difficult to adhere to a calorie restriction (CR), the most crucial step for weight loss, because calorie intake must be carefully monitored every day. Ultimate success requires substantial commitment in addition to clear and solid recommendations and support from the treatment team, which might be organized at the community level. Important hurdles to weight loss such as medical comorbidities, low education levels, and limited access to healthy food should be considered when developing a treatment plan. While CR is a conventional recommendation to help people with MAFLD achieve their weight reduction goals, another optional approach that limits the timing of food intake has recently been introduced. Time-restricted eating may be an attractive alternative to CR for weight loss in patients with T2D as well as in MAFLD and can be utilized easily in the community [58,59]. Alternatively, despite surgical risk, bariatric surgery in MAFLD patients with severe obesity can lead to substantial (15–25%), durable weight reduction and improvement in liver histological features of MASH and fibrosis [60]. Weight loss improves MAFLD and all cardio-metabolic comorbidities, which then favorably affects cardiovascular and malignancy-related risk. There is an independent contribution of MASH to cardiovascular and cancer risk, but we do not know yet if liver targeted treatment interventions will reduce them.

## 9. Optimizing Management with Existing Therapeutics

Irrespective of numerous studies and activities to treat MAFLD, there is only one FDA-approved therapy for MASH now. A combination of conservative treatments (lifestyle adjustments, increasing physical activity and smoking/alcohol cessation) appear to be beneficial. Intriguingly, several drugs that are available for other indications have been studied for MAFLD treatment (Table 1). Similarly to the experiences with several hepatotonics, ursodeoxycholic acid, ω-3 fatty acids and metformin have not shown any histological benefit. Metformin, an important diabetic treatment medicine relieving IR has no effect on MASH disease activity [61]. Vitamin E and pioglitazone have been endorsed by current guidelines as possible treatment in patients with MASH [62,63,64,65]. The benefits of vitamin E for MASH have been proved in several studies of patients without diabetes or cirrhosis. In a trial of patients with T2D and MASH, combination therapy of vitamin E with pioglitazone achieved improvement of MASH without worsening of fibrosis [64]. Nonetheless, vitamin E can only be used taking potential adverse effects (increased risk of bleeding and adverse cardiovascular outcomes in higher doses) into account. Although statins have no discernible histological benefit on MASH, they should be used for cardiovascular risk reduction [66].

Thiazolidinediones (TZDs), to which pioglitazone belongs, activate PPARγ in adipose tissue and has been proposed to promote differentiation of adipocytes and storage of fat in adipose tissue, protecting peripheral tissues from lipotoxicity [76]. Fortunately, all TZDs are highly effective in improving MAFLD outcomes. Multiple trials in MASH patients with and without diabetes have shown that pioglitazone improves MASH activity with some improvement in fibrosis [76,77]. Weight gain and the risk of bone loss are side effects. Accordingly, it appears unlikely that either vitamin E or pioglitazone will be studied in phase 3 studies. Alternatively, other drugs that modulate PPAR-γ and complementary mechanisms are being developed. GLP-1 receptor agonists [68,69,78,79] and SGLT2 inhibitors [69,80,81], antidiabetic medications that are cardioprotective and renoprotective, are currently being studied in several trials to assess their efficacy in MASH resolution and improvement in fibrosis. These drugs have additional benefit of inducing weight loss. Several GLP-1 receptor agonists are being evaluated for the treatment of MASH with some success. In a phase 2 trial, s.c. semaglutide 0.4 mg daily proved MASH resolution without worsening of fibrosis [67]. Although it is difficult to discern if this effect is independent of weight loss, it shows the highest rate of MASH resolution ever reported in MASH therapeutic trials [80,81]. Interestingly, in a recent Korean study of T2D with MAFLD, SGLT2 inhibitors showed a higher MAFLD regression and lower adverse liver-related outcomes compared with other oral anti-diabetic drugs including TZDs [82].

## 10. Emerging Therapeutics of MASH

Numerous drugs for MASH treatment with different mechanisms of action, aiming to target lipid metabolism, inflammatory, or fibrotic pathways, are in development [83] (Figure 1). Trials utilizing potent FXR agonist have proven favorable results [70,84,85]. A trial of obeticholic acid [84] was implemented as the first phase 3 trial to prove an improvement in fibrosis without worsening of MASH, confirming the findings of the previous phase 2 trial [70,85]. However, the magnitude of response was modest, suggesting that combination therapy would be required to treat the majority of patients adequately. Although obeticholic acid failed to achieve complete MASH resolution, it did improve individual histological features. However, pruritus and increase in LDL concentration are side effects of obeticholic acid. Recently, the selective thyroid hormone receptor β (THR-β) agonists have been found to be effective in the treatment of MAFLD [72,73], and approved by FDA. In addition, irrespective of multiple failures caused by disease heterogeneity, several other drugs are in advanced stages of development for MASH [71,74,75]. However, future treatment will require combination therapy since the modest effect and potential side effects in higher doses of these drugs. Several combination trials are also now underway. The ATLAS trial showed an improvement in fibrosis with cilofexor (a FXR agonist) and firsocostat (an ACC inhibitor) in patients with MASH and fibrosis [86]. Patients receiving this combination were more likely to have better improvement in the MAFLD activity score than those receiving monotherapy. Given the modest difference, more effective combinations will be needed.

## 11. miRNA Targeting for MAFLD

MAFLD is associated with thorough reprogramming of hepatic metabolism. Research indicates that various metabolic pathways deregulated in MAFLD converge in the aberrant accumulation of lipids into hepatocytes. In recent years, microRNAs (miRs) have been proposed to halt the development and progression of MAFLD as alterations of epigenetic mechanisms may contribute to these hepatic metabolic changes [87]. As elevated glucose levels can lead to increased DNL, miR-dependent alterations of hepatic glycolysis, gluconeogenesis and glycogen metabolism are key pathological mechanisms contributing to MAFLD development. In addition, deregulated cellular processes, such as autophagy or ER stress and the UPR were also implicated in steatosis and shown to be under the control of several miRs. However, to develop an efficient miR-based therapeutic strategy, targeting common key transcription factors involved in hepatic glucose and lipid metabolism would be important to have a broader systemic effect on MAFLD [88]. MiR-192 and miR-29, which are both deregulated in the liver of patients with MAFLD, were reported to target SREBP1, while ChREBP is regulated by miR-1322 in hepatocytes. LXR, which can regulate the activity of both SREBP1 and ChREBP, was also found to be targeted by several miRs (miR-1, miR-155, miR-206, and miR-613). PPARγ, a lipogenic transcription factor, was found to be regulated by miR-27a, miR-34a, miR-128, and miR130. In addition, PPARγ is also involved in hepatic stellate cells activation repression, thereby negatively regulating hepatic fibrosis. MiR-696 and miR-130a are targeting its cofactor, PGC1α, while miR-16 and miR-100 are acting on its repressor, NCOR. Another strategy to alleviate steatosis in the liver is to activate lipid oxidation by derepressing expression of PPARα. PPARα was shown to be targeted by miR-9, miR-10b, miR-21, miR-33, and miR-199a. However, as all these transcription factors or cofactors are provisional potential targets to be regulated, which have been examined only in some cell lines and/ or in animal models, their real influence in MAFLD development and progression in man has not yet been determined.

## 12. Therapeutic Potential of a New Antioxidant Protein Delivery in MAFLD

Though diabetes and/or MAFLD progression are commonly associated with increased OS, only vitamin E, an important antioxidant, but not other antioxidants, has been shown to improve the histology of MAFLD. These inconsistent effects of antioxidants might be resulted from diverse factors such as difficulty in maintaining a consistent circulating antioxidant level, inadequate tissue distribution, and lack of suitable exogenous antioxidants. Therapeutic antioxidants should be given in appropriate cellular concentration enough to tackle newly generating ROS levels without significant toxicity [89]. Therefore, a different way of delivering medicines that have antioxidant properties has been studied [90]. If antioxidants, in general, are to improve the pathophysiology in various stages of MAFLD, steady delivery at required concentrations into the cells or into the cellular compartment where ROS are generated would be most important. A strategy to induce an endogenous and nonspecific antioxidant or to deliver it in a more efficient way could be a more attractive approach. As MAFLD progression might be resulted from OS in mitochondria, ER and others (especially in hepatocyte as well as stellate cells) due to lipotoxicity, hypoxia, and AGE accumulation, endogenous nonspecific antioxidant treatment may improve tissue repair in various stages of MAFLD. Therefore, nonspecific antioxidants such as metallothionein (MT), SOD, and catalase have been exploited to combat increased ROS in various tissues applying intracellular delivery [89,90,91,92,93]. Taking advantage of the recently developed cell-penetrating peptide technologies, each of these antioxidants and their combination inhibited various OS and had the potential to protect cells and tissues against MAFLD progression as well as diabetes and its complications due to its anti-apoptotic and antioxidant effects either in vitro or in vivo [93]. Tat-MT and Tat-SOD given in combination intraperitoneally at regular interval were protective against various injuries and protected against steatosis as well as inflammatory condition in OLETF rats (unpublished data). This result is reminiscent of a similar delay in development of MASH by vitamin E.

## 13. Challenges and Prospects

Although progress has been made during the past decades in investigating the natural history and underlying biology of MAFLD, there are still many areas of challenges and obstacles. MAFLD in general, and MASH in particular, are poorly understood and recognized by health care professionals working in the community, especially in Korea. The implementation of appropriate strategies to identify and manage at-risk patients with MAFLD and advanced fibrosis will require timely action from clinicians in community care settings, diabetes clinics, and other specialists who treat patients with metabolic risk factors. There is an increasing need for a multi-pronged public health approach to cope with increased MAFLD risk factors in rapidly modernizing communities. When considering liver fibrosis as a key prognostic marker for liver-related outcomes, it can be assessed through combinations of non-invasive tests in routine care of the patients in the community, helping overcome challenges in the obesogenic environment. In addition, several obstacles should be addressed in the development of highly effective therapeutic measures of intervention. One of the most important challenges in the community hospitals is the long-term need for liver biopsy for diagnosis and follow-up. Although reliable biomarkers that can accurately diagnose and stage MAFLD across the entire disease spectrum do not yet exist, a diagnostic biomarker, in conjunction with a prognostic biomarker, would allow for the identification of high-risk individuals on whom resources should be concentrated. A second challenge is the substantial heterogeneity of MAFLD and the current limited understanding of disease phenotypes. The ability to phenotype patients would enable more accurate prognostication, better selection of appropriate therapy, and prediction of treatment response than is currently possible. Lastly, the refinement of therapeutic strategies into thoughtful combination approaches, tailored to the patient’s individual disease inducers, is needed for increased response rates and a change in our attitude toward screening. Finally, regardless of the progress that has been, or will be, made in diagnostic tests and drug treatments, healthy lifestyle and weight reduction remain crucial for the prevention and treatment of MAFLD, as obesity is the main driver of this common liver disease and its associated metabolic comorbidities.

## Figures and Tables

**Figure 1 ijms-26-02758-f001:**
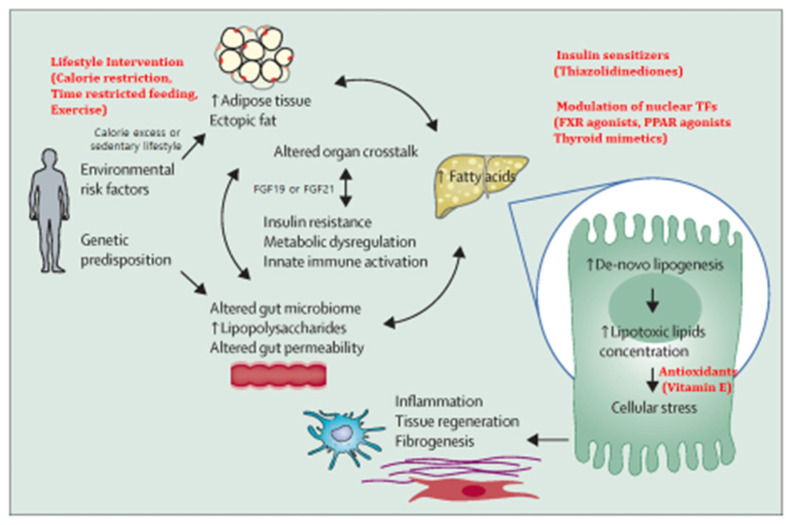
Schematic presentation of pathogenesis of MAFLD and possible mechanisms of action of available treatment strategies. The imbalance in lipid metabolism due to overnutrition leads to the formation of lipotoxic lipids that contribute to oxidative and endoplasmic reticulum stress, inflammasome activation and apoptotic cell death, and stimulation of inflammation, tissue regeneration, and fibrogenesis. The pathogenic pathways of MAFLD influenced by metabolic, genetic, and microbiome-related factors were also depicted. The crosstalk between the liver and other organs (particularly adipose tissue and the gut) via FGF19 and FGF21 might also contribute to metabolic dysregulation and inflammation in MAFLD.

**Figure 2 ijms-26-02758-f002:**
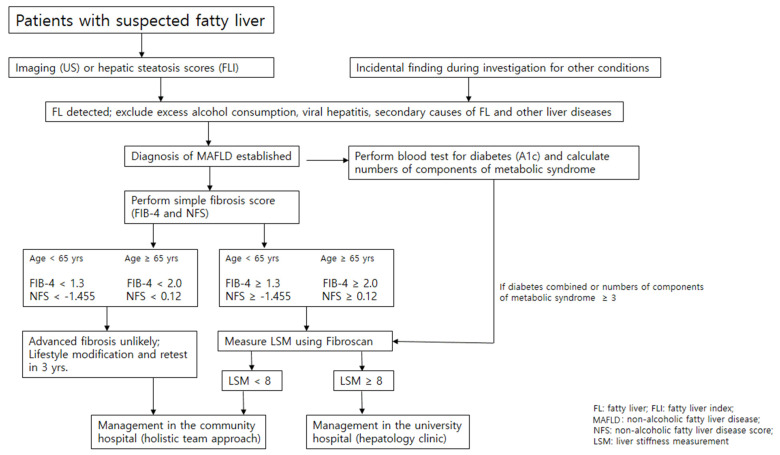
Consecutive assessment of the severity of MAFLD in the community. Inexpensive, simple fibrosis scores (NAFLD fibrosis score or FIB-4) can be counted as a first step to identify individuals at low risk of advanced fibrosis, who can be managed in the community. Individuals with high-risk scores require additional assessment with Fibro-scan or might require referral to university hospitals (secondary care) for investigation or management of advanced fibrosis. Patients without advanced fibrosis at initial assessment might require ongoing monitoring to identify progressive liver disease and retesting 3–5 years after the initial assessment.

**Table 1 ijms-26-02758-t001:** Comparison of landmark studies of different ways of management of MAFLD.

Treatment Modality	Specific Treatment	Subjects	Effect	Quality of Evidence Followed	Adverse Reactions	References
Lifestyle-induced weight loss	Weight reduction ≥ 7%	MAFLD (MASH)	Improve disease severity	Several small to moderate RCTs	None	[57]
Bariatric surgery	Roux-Y Gastric Bypass	MASH	Resolve MAFLD/MASH and regress fibrosis	Several small to moderate RCTs	Not completely safe due to surgery	[60]
Metformin	Metformin	MASH	Improve ALT, did not improve MASH histology	None, because of clear information	Hypoglycemia, lactic acidosis	[61]
Thiazolidinedione	Pioglitazone	MASH (most are non-diabetics)	Improve MAFLD activity score, resolve MASH in some patients		Edema, bone loss	[62]
	Pioglitazone	MASH (prediabetes, or T2D)	Improve steatosis, inflammation and fibrosis progression, but did not improve fibrosis	Several small to moderate phase 2 RCTs	Edema, bone loss	[61,63]
Statins	Simvastatin	MASH	Not improve steatosis, fibrosis, necroinflammation	None, because of clear information		[61,66]
Antioxidants	Vitamin E (800 IU/D, 2 yrs)	MASH	Improve MAFLD acitivity score, resolve definite MASH in some patients	Several small to moderate RCTs	Bleeding, adverse CV outcomes in higher doses	[62,64,65]
GLP-1 agonists	Semaglutide 0.4 mg/D 72 wks	MASH	Improves hepatic steatosis, necroinflammation	Several small to moderate RCTs	Safe and effective (except nausea, anorexia, constipation)	[67,68]
SGLT2 inhibitors	Dapagliflozin 10 mg/D 12 wks	MAFLD with T2D	Improves hepatic steatosis, necroinflammation, and liver enzymes	Several small RCTs with non-invasive tests	hypoglycemia, ketoacidosis, urinary tract and genital infections	[69]
FXR agonists	Obeticholic acid 25 mg/D 72 wks	MASH	Improve histology of MASH	Improve liver fibrosis in a phase III RCT	severe adverse reactions (pruritus, LDL increase) limit long-term use	[70]
PPARα agonists	Saroglitazar 4 mg/D 16 wks	MASH	improve dyslipidemia; reduce fat and triglyceride in the liver		Safe and effective	[71]
THRβ agonist	Resmetirom 80 mg/D 36 wks	MASH	Reduce liver fat content	FDA approved	Safe and effective, but further studies are needed	[72,73]
FGF19 analog	NGM282 1 or 3 mg/D 12 wks	MASH	improve histological features and fibrosis score of MASH	Well-tolerated in a phase IIb study	Cholesterol increase, nausea, vomiting, diarrhea.	[74]
FGF21 analog	Pegozafermin s.c. weeky of biweekly	MASH	improve histological features and fibrosis score of MASH	Well-tolerated in a phase Ib/IIa study	Safe and effective	[75]

Abbreviations; MAFLD: nonalcoholic fatty liver disease; MASH: nonalcoholic steatohepatitis; RCT: randomized controlled trial; ALT: alanine aminotransferase; T2D: type 2 diabetes; CV: cardiovascular.

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
