# Peer review of "Non-Alcoholic Fatty Liver Disease (NAFLD) Management in the Community"

_ijms, 2025, doi:10.3390/ijms26062758_

Round 1
Reviewer 1 Report
Comments and Suggestions for Authors
Thank you for your submission on “Non-Alcoholic Fatty Liver Disease (NAFLD) Management in the Community.”Your manuscript provides a comprehensive review of NAFLD, covering its epidemiology, pathogenesis, diagnostic strategies, and management approaches, with a particular focus on community-based care in Korea. The review is well-structured and informative, offering valuable insights into non-invasive diagnostics and emerging therapies.
However, there are several areas where improvements can be made to enhance the clarity, scientific depth, and readability of the manuscript. Enclosed are specific comments and suggestions to strengthen the paper.

its good but the authors can follow the suggestion and improve
Author Response
Reviewer 1
Summary
Thank you very much for taking the time to review this manuscript. Please find the detailed responses below and the corresponding revisions/corrections highlighted/in track changes in the re-submitted files.
- It’s very well accepted in the field that NAFLD be now addressed as MAFLD, the reviewers have mentioned it in the literature but have not adhered to the guideline, I strongly suggest the reviewers to follow the guideline and replace NAFLD to MAFLD and NASH to MASH, until a strong reason is provided to not do so.
Thank you for your suggestion. The authors agreed to follow your suggestion and replaced the term ’NAFLD’ to ‘MAFLD’ and ‘NASH’ to ‘MASH’ after we introduced the term, ‘MAFLD’ in the manuscript.
In fact, in 2020, an international panel of experts proposed the concept of MAFLD to emphasize the contribution of metabolic risk factors to the development and progression of liver disease (even among patients with other liver diseases) (J Hepatol 73: 202–09, 2020). However, MAFLD is not the accepted nomenclature by the American Association for the Study of Liver Diseases or the European Association for the Study of Liver Diseases. (Lancet 397: 2212–24,2021; Hepatology published online June 16, 2020. https://doi.org/10.1002/hep.31420). However, we prefer to use MAFLD and MASH rather than NAFLD and NASH, because of the concise diagnostic criteria, removal of the requirement to exclude concomitant liver diseases, and reduction in the stigma associated with the condition.
- The manuscript does not differentiate between lean NAFLD vs. obesity-related NAFLD, which is particularly relevant in Korea where lean NAFLD is more prevalent. The reviewer would like to suggest to include a discussion on lean NAFLD and its unique pathophysiology.
Thank you for your suggestion. We agreed to follow your suggestion and added a discussion on ‘lean NAFLD’ in the ‘Introduction’ of the manuscript to introduce the difference between lean NAFLD vs. obesity-related NAFLD. We also add some candidate gene lists of lean NAFLD after we introduced genetic markers of NAFLD and genetic risk stratification in the manuscript.
- The authors have very briefly discussed genetic risk stratification and the potential role of microbiome-based therapies, to which the reviewer would like to suggest to be a bit more detailed
Thank you for your suggestion. We agreed to follow your suggestion and added following discussion on ‘genetic risk stratification’ in the manuscript.
In addition to elevated caloric intake and a sedentary lifestyle, genetic and epigenetic predisposition contribute to the development of MAFLD. Utilizing genetic markers found in genome-wide association studies (GWAS), functional studies in rodents and epigenetic markers, researchers tried to predict development of MAFLD and MASH. In an ethnically diverse cohort, one group replicated several key genetic variants for MAFLD and showed the utility of genetic risk stratification (GRS) for MAFLD risk prediction (Hepatol Commun 5: 1689-1703, 2021). However, applicability of currently defined risk scores for risk stratification is to be concluded. Further efforts are needed to make the scores more usable and meaningful.
Thank you again for your suggestion. We agreed to follow your suggestion and added a discussion on gut microbiome signature and elaborated on the potential role of microbiome-based therapies.
.
The composition of bacteria in the gastrointestinal tract can enhance fat deposition, modulate energy metabolism and alter inflammatory processes. Therefore, alterations in gut microbiota composition are seen in patients with MAFLD. Emerging evidence suggests a role for the gut microbiome in obesity/ metabolic syndrome prediction and progress in understanding the association of the gut microbiome signature with NAFLD disease severity (Clin Chim Acta 523: 304-314, 2021; Cell Metab 25: 1054–62.e5, 2017; Nat Commun 10: 1406, 2019). Gut microbiota plays a role in the pathophysiology of MAFLD through the gut-liver axis. Microbiome-based therapies (probiotics, prebiotics, synbiotic, fecal microbiota transfer, polyphenols, specific diets, and exercise interventions) have been found to modify gut microbiota signatures and improve MAFLD outcomes. Apart from probiotics that have already been tested in human RCTs, most of these potential therapeutics have been studied in animals. Their efficacy still warrants confirmation in humans.
Although preclinical studies indicate that probiotic and postbiotic administration improves different liver markers, no data obtained in humans have been published so far since all the studies are ongoing clinical trials. We might expect cautiously that microbiome-based therapy is safe and effective for the management of MAFLD (Crit Rev Food Sci Nutr 7: 1-24, 2024).
Reviewer 2 Report
Comments and Suggestions for Authors
This is a well-written review article focusing on the growing problems of T2DM and NAFLD/MAFLD in the Korean population. The review is comprehensive and easy to follow. However, there are a few concepts or points that could be enriched.
- The rise in T2DM and NAFLD since the late 1960s is a global problem related in part to excessive nutritional intake. It would be useful to comment on any obvious shifts in the dietary composition such as reduced pulses/legumes/tofu, increased rice and Western starchy carbohydrate intake, to account for the epidemiologic shifts.
- The nature of dyslipidemia could be expanded. Is anything known about LDL, ApoB, LP(a), etc in the population?
- High rates of T2DM and NAFLD in what has been termed “skinny fat” are problematic due to lack of obvious features that might trigger suspicion. What might be the relevance of measuring waist circumference and routinely assessing visceral obesity? Is there data focusing on the management of visceral/abdominal adiposity to reduce NAFLD?
- Please provide any updated GLP-1 receptor agonist clinical trial data related to NAFLD from any country, as this is a rapidly moving target.
Author Response
Reviewer 2:
Summary
Thank you very much for taking the time to review this manuscript. Please find the detailed responses below and the corresponding revisions/corrections highlighted/in track changes in the re-submitted files.
- The rise in T2DM and NAFLD since the late 1960s is a global problem related in part to excessive nutritional intake. It would be useful to comment on any obvious shifts in the dietary composition such as reduced pulses/legumes/tofu, increased rice and Western starchy carbohydrate intake, to account for the epidemiologic shifts.
Thank you for your suggestion. The authors agreed to follow your suggestion and added a brief discussion on the shifts in the dietary composition and influence of Westernization to account for the epidemiologic changes in Korea.
The rise in T2D and NAFLD since the late 1970s might be resulted in part from excessive nutritional intake. When we think this increase in T2D and NAFLD is associated with the surge of metabolic constellation, however, It would be useful to assume that it might also be related with shift in the dietary composition taking tremendous changes of basic nutrient composition of diet into account. There has been obvious shifts in the dietary composition. According to the Korean Annual National Health and Nutrition Examination Survey, carbohydrate intake has been reduced gradually, whereas lipid intake has been increased much suggesting a role of dyslipidemia in this epidemiologic shift.
The rise in diabetes and MAFLD since the late 1960s might be resulted in part from excessive nutritional intake. However, it might also be related to shifts in the dietary composition taking tremendous changes of basic nutrient composition of diet into account. According to the Korean Annual National Health and Nutrition Examination Survey, carbohydrate intake has been reduced gradually, whereas lipid intake has been increased much suggesting a role of dyslipidemia in this epidemiologic shift (13).
- The nature of dyslipidemia could be expanded. Is anything known about LDL, ApoB, LP(a), etc in the population?
Thank you for your suggestion. We agreed to follow your suggestion and added a brief discussion on the nature of dyslipidemia, especially concentrating on LDL, triglyceride and ApoB.
Owing to shifts in the dietary composition as well as excessive nutrient intake, there has been an aggregation of CV risk factors in the general population suggesting involvement of metabolic syndrome (MS).
In Yonchon county, Korea, there was an aggregation of CV risk factors in the general population even in 1992 suggesting involvement of metabolic syndrome (MS). There, we found an association between levels of serum triglyceride (or even serum total cholesterol and LDL cholesterol levels) and diabetes status and MS status. Although average serum total cholesterol and LDL concentration levels were not high, there was a good correlation between those in 1993 and 1995, suggesting involvement of genetic or shared environment. Apolipoprotein B (apoB) concentration, although we do not have the data in the Yonchon county, reflects the number of atherogenic particles and is closely associated with atherosclerosis, and better than non-HDL cholesterol in defining MS.
Although in a prospective study of 25,193 healthy Korean men without MS, serum ApoB levels were found to predict MS, independent of non-HDL-cholesterol (Atherosclerosis 226: 496-501, 2013), no population-based studies of characteristic proatherogenic lipid profile such as apoB or LP (a) distribution has been performed in the community, which makes these field beyond our focus in the manuscript. One laboratory based study firstly intended to show the prevalence of various types of dyslipoproteinemias in Korean population. The percentile values of Korean population were similar to those of NHANES. Integration of lipid markers is needed for making clinical decisions and further research involving various populations and methodologies should be performed.
These might be resulted from increased cardiometabolic risk factors that are shared in MAFLD and CVDs. However, it is not known to what extent MAFLD has a direct causative role in the development of CVDs. In addition to the characteristic proatherogenic lipid profile, the bidirectional relationship between MAFLD and some MS features (T2D and hypertension) is one mechanism by which MAFLD might augment cardiovascular risk (46, 47).
- High rates of T2DM and NAFLD in what has been termed “skinny fat” are problematic due to lack of obvious features that might trigger suspicion. What might be the relevance of measuring waist circumference and routinely assessing visceral obesity? Is there data focusing on the management of visceral/abdominal adiposity to reduce NAFLD?
Thank you for your suggestion. In fact, we are measuring waist circumference (WC) routinely in the first visit of ‘would-be NAFLD’ to assess severity by way of counting numbers of components of MS involved. However, we do not have the data on the management of visceral/abdominal adiposity to reduce NAFLD yet.
Some people with a healthy BMI (<23kg/m2 in Asians) can still develop NAFLD, often described as non-obese or lean NAFLD. These patients usually have central obesity or other metabolic risk factors (Am J Gastroenterol 110: 1306–14, 2015). It would be a good idea to measure waist circumference (WC) routinely in addition to BMI to manage visceral adiposity to reduce NAFLD or type 2 diabetes. We suggest that numbers of components of MS involved should be counted at first visit taking the importance of routine WC measurement into account and could be used as an important prognostic information of NAFLD assessed easily in the routine care of NAFLD in the manuscript. This way, we can find a way to modulate visceral/abdominal adiposity independent of BMI to reduce NAFLD. In Yonchon County, we found an independent association of abdominal obesity as well as obesity with T2D prevalence (Diabetes Care 18: 534-538, 1995). A cross-sectional National Health Examination Survey carried out in 1998, also found the similar association between clustering of CVD risk factors and BMI or WC (Cardiovascular Risk Factors 6/6: 335-344, 1996; Asia Pac J Clin Nutr 12: 411-8, 2003). However, we have not studied the relative contribution of WC on the development of NAFLD properly yet.
- Please provide any updated GLP-1 receptor agonist clinical trial data related to NAFLD from any country, as this is a rapidly moving target.
We already included the clinical trial data of GLP-1 receptor agonists in the manuscript. However, none of the any other updated trial data from any country have been published so far since all the studies are ongoing clinical trials.
Several RCTs found that GLP-1 receptor agonists, semaglutide and liraglutide, may also be effective in resolution of MASH but not of fibrosis. Eight randomized controlled trials on T2D and NAFLD from the Cochrane Library, Embase, and PubMed in one meta-analysis found that in patients with T2D and NAFLD which presents a strong signal that GLP-1 receptor agonists improve liver function and histology by improving glycemia, reducing body weight and hepatic fat, which in turn reduces hepatic inflammation. Preliminary data from interventions with tirzepatide, a dual GLP-1 and glucose-dependent insulinotropic polypeptide RA are encouraging, but more data based on liver biopsy are needed.